# Reduced Cross-Shift Lung Function and Respiratory Symptoms among Integrated Textile Factory Workers in Ethiopia

**DOI:** 10.3390/ijerph17082741

**Published:** 2020-04-16

**Authors:** Yifokire Tefera Zele, Abera Kumie, Wakgari Deressa, Bente E. Moen, Magne Bråtveit

**Affiliations:** 1Department of Preventive Medicine, School of Public Health, College of Health Sciences, Addis Ababa University, P.O. Box 9086 Addis Ababa, Ethiopia; aberakumie2@yahoo.com (A.K.); deressaw@gmail.com (W.D.); 2Department of Global Public Health and Primary Care, University of Bergen, 5007 Bergen, Norway; Bente.Moen@uib.no (B.E.M.); Magne.Bratveit@uib.no (M.B.); 3Centre for International Health, Department of Global Public Health and Primary Care, Faculty of Medicine, University of Bergen, Årstadveien 21, 5009 Bergen, Norway

**Keywords:** cross-shift, lung function, chronic respiratory symptoms, integrated textile factory

## Abstract

Chronic respiratory symptoms and reduction in lung function has been described as a common health problem among textile workers in low- and middle-income countries. The objective of this study was to measure lung function and respiratory symptoms among workers from an integrated textile factory. A comparative cross-sectional study design with a cross-shift lung function measurement was performed in 306 cotton dust exposed workers from an integrated textile factory and 156 control workers from a water bottling factory. An integrated textile factory typically has four main production departments (spinning, weaving, finishing, and garment) that process raw cotton and manufacture clothes or fabrics. Respiratory symptoms were assessed by adopting the standard American Thoracic Society questionnaire. Descriptive statistics and logistic and linear regression analysis were used. The prevalence of respiratory symptoms was significantly higher among textile workers (54%) than in controls (28%). Chronic cough, chest tightness, and breathlessness were significantly higher among textile workers (23%, 33%, and 37%, respectively) than in the control group (5%, 17% and 6%, respectively). Breathlessness was the most prevalent chronic respiratory symptom with highest adjusted odds ratio 9.4 (95% CI 4.4–20.3). A significantly higher cross-shift lung function reduction was observed among textile workers (123 mL for FEV_1_ and 129 mL for FVC) compared with the control group (14 mL for FEV_1_ and 12 mL for FVC). Thus, workers’ respiratory health protection programs should be strengthened in textile factories.

## 1. Introduction

The textile and clothing industries are major sources for economic growth and social development in developing countries. In some low-income countries, the sector accounts for up to 15% of the Gross Domestic Product (GDP) and creates employment for 35%–90% of the total number of workers in manufacturing industries [1]. Furthermore, the textile sector is one of the key components of the development agenda in the Ethiopian Growth and Transformation Plan and is expected to create increasing job opportunities [2].

However, several studies have reported that workers in the textile industry have been vulnerable to and suffer from respiratory health problems [3,4,5,6]. A recent study from a textile factory in Ethiopia reported a high prevalence of chronic respiratory symptoms including cough (64.7%), chest tightness (43%), breathlessness (41%), and wheezing (39%) [7]. Studies in other parts of the world have also reported a higher prevalence of respiratory symptoms among textile workers than among controls [8,9,10,11]. Various studies [8,12,13,14] have shown high exposure levels of dust and endotoxin in the textile factories, and the dust levels have been related to the development of respiratory symptoms [5,12,15].

Interviews about symptoms have limitations, and there is a need for objective studies of these respiratory problems. It is of interest to study an objective measure—for instance, whether there is a change in lung function after a working day in the textile industry by conducting cross-shift lung function measurements.

Cross-shift reduction in lung function has previously been reported among textile workers in other countries [15,16]. In a Shanghai textile cohort, the result of three surveys indicated a larger cross-shift ∆FEV1 decrease among exposed cotton workers (48–58 mL) compared to controls (6–26 mL) [17]. Some studies in the cotton processing textile industry recorded more pronounced cross-shift lung function reduction among workers in high dust-exposed sections compared to low-dust level sections [15,18]. Earlier cross-shift studies in an Ethiopian textile factory have also shown acute lung function reduction [19,20]. However, one study involved only workers from the spinning and weaving departments and did not have a control group [20].

There have been changes in the textile industry in Ethiopia and other developing countries, such as renewal of machineries, installation of ventilation systems, more females in the work force, and workers moving across the departments [6]. In Ethiopia, textile factories have been encouraged to adopt an integrated production model. An integrated textile factory typically has four main production departments (spinning, weaving, finishing, and garment) that process raw cotton and manufacture clothes or fabrics. The major tasks or activities in each of these departments are the production of yarn (spinning), the production of fabric (weaving), ensuring the proper appearance of the fabric (finishing), and cutting and sewing to make clothing (garment). We have no previous studies from these integrated factories. It is therefore important to examine Ethiopian textile workers’ current respiratory health status using both subjective and objective measures.

The aim of this study was to measure the cross-shift lung function and respiratory symptoms among workers in an integrated textile factory compared to a control group of workers with low exposure to dust. The findings of the study might help to ascertain the risk of adverse respiratory health among workers in the Ethiopian textile industry today.

## 2. Methods and Materials

### 2.1. Study Design, Study Setting, and Period

A comparative cross-sectional study of respiratory symptoms and cross-shift lung function measurements were conducted from November 2017 to January 2018 among workers in an integrated textile factory and among control workers from two water bottling and soft drink factories.

### 2.2. Exposed Workers

Exposed workers were selected from an integrated textile factory situated 550 km northwest of the capital city, Addis Ababa. Although the factory was established in 1961 and is one of the oldest textile factories in Ethiopia, changes were made in 2011, such as replacement of machinery and improvements in the ventilation system. The factory has four departments—spinning, weaving, finishing, and garment. These departments are located in separate but interconnected sections in one building, except for the garment department, which is located in a separate building. The factory has a three-shift work pattern: day shift (07:00–15:00), evening shift (15:00–23:00) and night shift (23:00–07:00). The shift rotates every week with a shift cycle of three weeks.

### 2.3. Control Group

Controls were selected from two soft drink and mineral water bottling factories. Such factories are expected to have low dust exposure level and have been used as controls in studies of respiratory health in other industries [21,22]. One of the factories is located about 180 km from the textile factory, and the other factory is situated in the same town as the textile factory. The first of these factories has two shifts: day shift (07:00–15:00) and evening shift (15:00–23:00) with a shift rotation of every week, and the second one has only a day shifts (08:00–16:00).

### 2.4. Sample Size and Sampling Techniques

The sample size for this study was calculated by the double population formula using prevalence of respiratory health symptom among exposed cotton workers (21%) and controls (8.4%) [23]. Applying 90% power, 0.05 significance level, a ratio of 2:1 for exposed vs control, and adding 10% for non-respondents, we needed 306 exposed workers from the integrated textile factory and 156 control workers from the soft drink and water bottling factories. For the lung function study, a total sample of 379 subjects was calculated based on a previous study [17]—251 workers from the textile factory and 128 workers from the beverage bottling factory.

In all, 1136 production workers in the integrated textile factory and 750 workers in the control factories were available. A total of 462 workers (306 from the integrated textile factory and 156 from soft drink and water bottling factory) were selected for respiratory symptoms interviews and lung function spirometer measurements. The textile factory had four departments and within each department, a random sampling technique was used to select participants from a list of workers obtained from the human resource office at the factory. Sampled workers from the integrated textile factory departments were 81 out of 298 from spinning, 103 out of 380 from weaving, 52 out of 183 from finishing, and 70 out of 260 from garment. The controls were selected from both the production and administrative workers. A sample of 44 workers from the administration and 112 workers from the production personnel were selected randomly using the list of workers obtained from the human resource personnel (Figure 1).

### 2.5. Data Collection

#### 2.5.1. Socio-Demographic Data Collection

Workers responded to questions about their sociodemographic and individual characteristics by answering a structured interview. These characteristics were age, sex, height, weight and educational level (primary, secondary, college and above), occupational history (total year of service in similar sector, year of service in current department and year of service in other departments), working department and job, use of face mask (yes/no), cooking own food (yes/no) use of biomass fuel (yes/no), and living with animals (yes/no). Participants were asked about their smoking habits—have you ever smoked (yes/no), currently smoke (yes/no), and number and years of smoking. Ever-smokers were participants who smoke currently or had smoked more than 20 packs of cigarettes during their lifetime or more than one cigarette a day for one year.

#### 2.5.2. Chronic Respiratory Symptoms

Respiratory symptoms were assessed through face-to-face interviews using the American Thoracic Society’s ATS-DLD-78-A standardized questionnaire [24]. Four trained nurses conducted the interviews in a separate room in the factory clinic using a questionnaire that had been translated into the local language and pre-tested. The questionnaire included queries about the presence of chronic symptoms of cough, chest tightness, breathlessness and wheezing (yes/no), and previous history of respiratory health problem (pneumonia, tuberculosis, bronchitis, asthma, and chest injury).

Cough—Participants were considered to have cough symptoms if they answered “yes” to at least one of the following four question: Do you usually cough first thing in the morning, cough during the day or night, cough as much as four to six times a day in a week, or cough most days during a period of three consecutive months during the year?

Chest tightness—Participants were considered to have chest tightness if he/she answered yes to the following question: Do you usually experience chest tightness while at work or just after work?

Breathlessness—Participants were considered to have breathlessness if he/she answered yes to the question: Do you usually troubled by a shortness of breath when walking hurriedly on level ground or walking up a slight hill, or get shortness of breath when walking at your own pace on level ground?

Wheezing—Participants were considered to have wheezing if he/she answered yes to the question: Does your chest ever sound wheezy or emit a whistling sound?

#### 2.5.3. Cross-Shift Lung Function

Cross-shift lung function measurements were performed using a computer-connected portable spirometer (SPIRARE 3 sensor model SPS 320) in accordance with the ATS recommendations [25]. Height and weight of the participants were measured with an instrument approved by the Ethiopian Standard Agency.

Three trained spirometer operators (physician, nurse and the investigator) were involved in the test. A step-wise spirometer measurement protocol based on the ATS was prepared for use in field data collection. All spirometer tests were performed on participants in the sitting posture. Three acceptable maneuvers with consistent (“repeatable”) results were retained, and the best values for Forced Vital Capacity (FVC) and Forced Expiratory Volume for one second (FEV1) were recorded.

Only the day and evening shift workers participated in this study. The day of the week taken off from the factory was not uniform among the workers; each worker’s day off in a week was from Monday to Sunday, depending on their preference. Hence, the spirometer was performed on all days of the week. Each worker was examined before and after work. For the day shift workers, the before shift and after shift spirometer measurements were taken between 07:00–08:00 and 15:00–16:00, respectively. Similarly, for the evening shift, the before shift and after shift spirometer measurements were taken between 15:00–16:00 and 22:00–23:00, respectively.

#### 2.5.4. Inhalable Dust and Endotoxin Exposure

In a previous study [26], personal inhalable dust was sampled in January and February 2017. In total, 64 workers were selected from four departments of the integrated textile factory for repeated sampling of inhalable dust (*n* = 96).

### 2.6. Data Management and Analysis

The principal investigator checked the collected data on site for completeness and consistency. Questionnaire data were entered in an epidemiological information package (Epi-Info) version 7.1 (Centers for Disease Control and Prevention, Atlanta, GA, USA), developed by CDC of the US whereas lung function test data were entered into an Excel spreadsheet and then exported to SPSS version 22 (IBM, Armonk, NY, USA) for data cleaning and analysis. The absolute values of the spirometer test measurements were used for the analysis of cross-shift change. The Global Lung Initiative Quanjer GLI-2012 multi-ethnic reference value for the African American ethnicity was used to estimate the predicted value and the proportion of subjects with FEV1 and FVC below the Lower Limit of Normal (LLN) [27].

The cross-shift changes in FEV1 (∆FEV1) and FVC (∆FVC) were calculated accordingly:
ΔFEV1 (mL)=FEV1(pre shift)−FEV1(post shift)ΔFVC (mL)=FVC (pre shift)−FVC(post shift)

Paired samples *t*-test, independent *t*-test, and ANOVA were used to compare the mean of continuous values within group and among groups, respectively. Pearson chi-square and Fisher exact tests were used to examine the difference among groups for the categorical variables. Binary logistic regression was used to determine the odds ratio of the respiratory symptoms when comparing exposed and control workers. The effect of textile exposure on the prevalence of respiratory symptoms was described by adjusted odds ratio (AOR) by adjusting the variables age, gender, education, biomass fuel use, animals living in the house, and working hours per week). Since the dust exposure varied among the textile departments, respiratory symptoms were analyzed by comparing each department with the garment department, as this had the lowest exposure levels.

Multiple linear regressions were used to estimate the difference in cross-shift lung function parameters between textile workers and controls when adjusting for age, sex, work shift, use of biomass fuel, and animals living in the house. Due to the possible diurnal change in the lung function, work shift was adjusted during analysis [28].

### 2.7. Ethical Approval

Ethical approval was obtained from the Institutional Review Board of Addis Ababa University College of Health Sciences (Protocol number: 057/16/SPH). An official request letter was sent from the Addis Ababa University School of Public Health to the factory and was discussed at meetings between the research investigator and the factory managers. The factory’s management granted access to the factories. Written informed consent was obtained from all study participants, and participation in the study was voluntary.

## 3. Results

### 3.1. Socio-Demographic Characteristics of Study Participants

A total of 458 subjects—303 textile workers and 155 controls from the soft drink and water bottling factories—participated, yielding a response rate of 99%. Four workers declined to participate (three of the textile workers and one from the control group). There was no statistical difference between exposed and controls concerning sex, age, height, service years, cooking own food, and previous respiratory illness. The workers from the control group had a higher educational level, used biomass fuel, and worked more hours per week than the textile workers. More textile workers had animals living in their house compared to the controls. There were only seven ever-smokers in the study—five in the exposed and two in the control group. Only two workers from the textile were current smokers. None of the participants used respiratory protective devices (face masks) (Table 1).

### 3.2. Respiratory Symptoms

The prevalence of chronic respiratory symptoms was higher among the textile workers (range 20%–37%) compared to controls (range 5%–17%). Breathlessness was the most prevalent chronic respiratory symptom (37%) among textile workers and had an odds ratio relative to controls of (9.4) when adjusted for the variables age, gender, education, biomass fuel, animals living in the house and working hours per week (Table 2). Generally, 54% of the textile workers and 28% of the controls reported having at least one chronic respiratory health symptom (Table 2).

Workers in the finishing department reported the highest prevalence of cough and breathlessness. These workers had significantly higher prevalence of cough than the garment workers, (AOR = 9.3, 95% CI: 2.8, 31.0), breathlessness (AOR = 2.64, 95% CI: 1.0, 6.6) and of at least one respiratory symptom (AOR = 3.4, 95% CI: 1.3, 8.5) adjusted for the variables age, gender, education, biomass fuel, animals living in the house and working hours per week. When adjusting for the same factors, workers in spinning and weaving had significantly higher prevalence than garment workers of chest tightness (AOR = 2.7, 95% CI: 1.3, 5.8) and cough (AOR = 3.3, 95% CI: 1.0, 10.4), respectively. The finishing department workers were older and had served more years in textile than workers in the spinning and weaving department (Table 3). The majority (84.6%) of the finishing department workers had previously worked in other departments.

### 3.3. Cross-Shift Lung Function

The predicted FEV1 and FVC value in the total population (exposed and control) was 3017 mL and 3592 mL, respectively. The proportion of subjects below the LLN value for the before shift FEV1 and FVC among the exposed group was 13% and 9%, respectively; among the control group, it was 9% and 6%, respectively, but the differences between the groups were not significant. About 4.4% of the participants had FEV1/FVC <70%; among exposed (5.4%) and among control (2.0%), however, the difference was not significant.

In the exposed group, there was a significant reduction in both FEV1 (123 mL) and FVC (129 mL) across the work shift (*p* < 0.001) (Table 4). In the control group, the across-work shift change for FEV1 (14 mL) or FVC (12 mL) was not significant. Workers in all departments in the textile factory except the finishing department had a significant across-shift decrease in spirometry for both FEV1 and FVC. The highest and the lowest across-shift change for FEV1 and FVC were in the spinning and finishing departments respectively (Table 4). There was no difference in cross-shift change FVC comparing the textile factory’s finishing department and the control.

The across-shift change in both FEV1 and FVC was significantly higher among the exposed workers compared to controls.

In the multiple linear regression analysis, the cross-shift change for both FEV1 and FVC were significantly higher in the exposed compared to the control when adjusted for the variables age, sex, work shift, use of biomass fuel for cooking and animals living in the house (Table 5).

## 4. Discussion

In this study, the textile workers showed a higher prevalence of chronic cough, chest tightness, and breathlessness than the control workers. A larger reduction in lung function across the work shift was found among textile workers compared to the controls.

Breathlessness, chest tightness, and cough were the most prevalent respiratory symptoms among the textile workers in this study. These symptoms are known to occur among textile workers and correspond with the diagnosis of byssinosis [6]. This is also found in other recent studies of the textiles where these symptoms are reported to have high prevalence [4,5,7,8,9,10,11,12,13,14,15,16,17,18,19,20,21,22,23,24,25,26,27,28,29].

The overall prevalence of chronic respiratory symptoms in this study (54%) was comparable to previous research conducted in similar low- and middle-income settings such as in Ethiopia (48%) [30], Egypt (59%) [9], Nigeria (62%) [11], and Bangladesh (53%) [31]. However, the prevalence was higher than the study conducted in an Iranian textile factory (26%), where workers were relatively younger [16].

Several previous studies have reported a higher prevalence of respiratory symptoms and a higher level of dust exposure in the first working phase of textile manufacturing, which is spinning [20,32,33,34]. This leads many of the recent respiratory symptom studies in the textile industry to recruit study population from spinning or weaving or from both departments [4,7,9,30]. However, in the current study, the highest prevalence of chronic respiratory symptoms was reported among workers in the finishing department (71%). Studies conducted in textile factories have indicated that workers who have the longest period of service associated with chronic exposure have the highest risk of developing chronic respiratory health problems [4,5,15,30]. This has been confirmed in this study, where workers in the finishing department were the oldest and were senior to the spinning and weaving department workers.

The cross-shift changes in both FEV1 and FVC differed significantly between the two groups, with the exposed group having the largest decline in both variables. In our study, the mean cross-shift reduction was higher than in the previous studies of Nepalese textile workers, where ∆FEV1 and ∆FVC were 74 mL and 81 mL, respectively [15], the Shanghai cohort of all the four surveys (∆FEV1; 48–67 mL) [17], and a study in Portuguese spinning (∆FEV1 = 86 mL) and weaving departments (∆FEV1 = 5 mL) [18]. However, our study result for the cross-shift change of FEV1 was not significantly different from the above-mentioned studies because their 95% CI included our mean value, except for the study in Portuguese weaving, which was significantly lower than the current study. The higher mean cross-shift reduction in the current study could be because our measurements were taken both in the morning and afternoon shift workers while the Nepalese study included only day shifts. Previous studies have shown that reductions in FEV1 and FVC are larger across time periods corresponding to evening shifts compared to morning shifts [35,36,37,38]. Besides, the difference in socio-demographic characteristics of the study population and difference on exposure level could be the possible reason for this difference. Comparatively exposure to endotoxin concentration per milligram of dust in our study was higher than all the above workplace measurements (Nepalese, Portuguese, and Shanghai cohorts) [13,18,26,39].

Only a small proportion of the workers had FEV1 and FVC values below the lower limit normal, suggesting that the majority of the participants had normal lung function. However, the predicted values have weakness as they are based on African Americans and not on Ethiopians, who may differ in socio-economic and demographic characteristics. On the other hand, less than 5% of the total study population had FEV1/FVC <0.7, indicating a low prevalence of airway obstruction among the participants.

In this study, a higher cross shift reduction was found in the spinning and weaving department workers than among workers in finishing and garment. Correspondingly, exposure to endotoxin was higher in spinning (1560 EU·m^−3^) and weaving (1086 EU·m^−3^) than in garment (258 EU·m^−3^) and finishing (76 EU·m^−3^). However, the exposure to inhalable dust was not different between these departments—spinning (0.71 mg·m^−3^), weaving (0.78 mg·m^−3^), finishing (1.25 mg·m^−3^), and garment (0.46 mg·m^−3^). The highest average inhalable dust exposure level was recorded in the carding section in the spinning department [26]. In a similar study, a higher cross-shift reduction was reported in spinning [15,16]. However, in our study, a high prevalence of chronic respiratory symptoms, but with the lowest cross-shift reduction, was found in the finishing department. This could be due to movement of workers with persistent respiratory symptoms from more highly exposed departments [6,40]. One of the longest longitudinal studies of a Shanghai textile cohort also indicated a significant cross-shift measurement variation between surveys [17]. Therefore, cross-shift lung function reduction may not be a direct indicator of the chronic respiratory health symptoms.

Strengths of this research include the objective measurement of exposure, the use of certified laboratory for analysis, the presence of control, full cooperation from the factory’s management and a high response rate. However, the exposure measurements were relatively few, and no measurements were performed in control groups. In addition, the cross-shift was not repeatedly measured, and production rate was not measured. The workers may have been a source of bias, as the textile workers may have wanted to change workplace status, leading to over-reporting of respiratory symptoms. However, the cross-shift lung function changes were found in the same groups as the ones with high exposure level to endotoxins, supporting the validity of the data. On the other hand, some workers may have had some reservations about complaining about their factory during the interview. We have no reason to believe this was the case, as the confidentiality in the study for each worker was fully observed. Another weakness of the study was the design. As it was a cross sectional study design, no causality between dust exposure and respiratory health can be concluded with certainty.

## 5. Conclusions

The prevalence of chronic respiratory symptoms was higher among textile factory workers compared to controls, and the textile workers had a higher level of cross-shift lung function reduction. The findings might be related to endotoxin exposure in the textile factory, but the results must be interpreted with caution due to the cross-sectional design of the study. Generally, workers’ respiratory protection program and dust reduction methods should be strengthened in textile factories.

## Figures and Tables

**Figure 1 ijerph-17-02741-f001:**
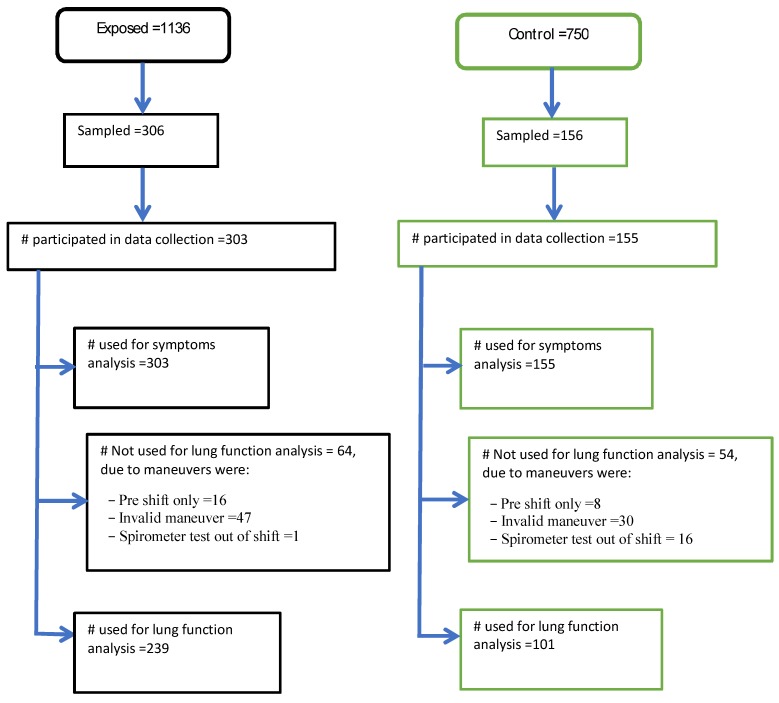
Schematic presentation of sample selection and analysis procedure for exposed and control.

**Table 1 ijerph-17-02741-t001:** Sociodemographic and exposure characteristics of study participants in the textile factory and among controls.

Variable	Textile Workers	Controls	*p*-Value
	N = 303	N = 155	
Age in years, AM (Range)	34 (22–63)	33 (19–62)	0.104 ^a^
Height in cm, AM (Range)	166 (144–186)	165 (149–185)	0.059 ^a^
Service in years, AM (Range)	10 (1–38)	9 (1–30)	0.138 ^a^
Body mass index, AM (Range)	22 (14–37)	22 (17–40)	0.238 ^a^
Sex:			
Female, *n* (%)	146 (48)	82 (53)	0.339 ^b^
Male, *n* (%)	157 (52)	73 (47)	
Education:			
Primary school, *n* (%)	59 (19)	24 (15)	0.038 ^b^
High school, *n* (%)	114 (38)	45 (29)	
College and above, *n* (%)	130 (43)	86 (55)	
Housing and living condition:			
Cook own food, *n* (%)	219 (72)	101 (65)	0.116 ^b^
Use biomass fuel, *n* (%)	198 (65)	123 (79)	0.002 ^b^
Ever-smokers, *n* (%)	5 (2)	2 (1)	1 ^c^
Animals living in the house, *n* (%)	57 (19)	15 (10)	0.011 ^b^
Previous respiratory illness, *n* (%)	13 (4)	2 (1)	0.088 ^b^
Working schedule:			
Working hours per week, AM (SD)	48 (1.4)	50 (6.4)	<0.001 ^a^
Day shift ^d^, *n* (%)	183 (60)	101 (65)	0.320 ^b^
Evening shift ^e^, *n* (%)	120 (40)	54 (35)	

^a^ Independent *t*-test; ^b^ Pearson chi square test; ^c^ Fisher’s exact test; ^d^ number of workers in day shift during lung function test; ^e^ number of workers in evening shift during lung function test AM = arithmetic mean; SD = standard deviation; N = number of workers in textile/control; *n* = number of workers within the group.

**Table 2 ijerph-17-02741-t002:** Prevalence of chronic respiratory symptoms among workers in the textile factory and among controls.

Variable	Textile Workers	Controls	^a^ Adjusted Odds Ratio (95% CI)
	(N = 303)	(N = 155)	
**Respiratory Symptoms**	***n* (%)**	***n* (%)**	
Cough	70 (23)	7 (5)	7.2 (3.1–17.0) **
Chest tightness	101 (33)	27 (17)	2.5 (1.5–4.2) **
Breathlessness	112 (37)	9 (6)	9.4 (4.4–20.3) **
Wheezing	60 (20)	20 (13)	1.6 (0.9–2.8) ^ns^
At least one respiratory symptom	163 (54)	43 (28)	3.1 (2.0–4.9) **

^a^ Logistic regression analysis adjusted for age, gender, education, biomass fuel, animals living in the house and working hours per week; ** = *p* < 0.001; ns = *p* > 0.05; N = number of workers in the groups; *n* = number of workers with symptom.

**Table 3 ijerph-17-02741-t003:** Workers age, service year, and respiratory symptoms across the different departments of the textile factory.

Departments	Age, Year	Service Years in Textile			Respiratory Symptoms				
		In Current Department	In Other Departments	Total	At least One Symptom	Cough	Chest Tightness	Breathlessness	Wheezing
	AM (SD)	AM (SD)	AM (SD)	AM (SD)	n (%)	n (%)	n (%)	n (%)	n (%)
Spinning, N = 81	33 (9)	6 (6)	2 (4)	8 (7)	42 (52) *^ns^*	12 (15) *^ns^*	34 (42) *	30 (37) ^*ns*^	15 (18) *^ns^*
Weaving, N = 101	33 (10)	8 (8)	1 (3)	10 (9)	54 (53) *^ns^*	31 (31) *	30 (30) ^*ns*^	31 (31) *^ns^*	21 (21) *^ns^*
Finishing, N = 52	37 (10)	7 (6)	7 (6)	13 (9)	37 (71) ***	21 (40) *	20 (38) *^ns^*	27 (52) *	9 (17) ^*ns*^
Garment, N = 69	35 (11)	5 (2)	6 (6)	11 (7)	30 (43) ^+^	6 (9) ^+^	17 (25) ^+^	24 (35) ^+^	15 (22) ^+^
*p*-value	0.040 ^a^	0.007 ^a^	<0.001 ^a^	0.024 ^a^	0.004 ^b^	<0.001 ^b^	0.001 ^b^	0.001 ^b^	0.413 ^b^

^a^*p*-value of ANOVA test; AM = arithmetic mean; (SD) = standard deviation; N = number of workers in the department; *n* = number of workers have symptom; ^b^
*p*-value for Pearson chi-square difference among departments; * and *^ns^* are significance level at *p*-value * *p* ≤ 0.05 and *^ns^ p* > 0.05 for logistic regression adjusted for age, gender, education, biomass fuel, animals living in the house and working hours per week; ^+^ Reference department because of low exposure level.

**Table 4 ijerph-17-02741-t004:** Comparison of before and after shift and cross-shift lung function parameters between the textile factory workers and controls.

Variable	*n*	FEV1, mL	FVC, mL	∆FEV1, mL		∆FVC, mL	
		Before Shift	Before Shift				
		AM (SD) ^a^	AM (SD) ^a^	AM(SD)	*p*-Value ^b^	AM (SD)	*p*-Value ^b^
Control	101	2910(637) ^ns^	3567(781) ^ns^	14(160)	*	12(174)	*
Textile	239	2999(715) **	3736(865) **	123(207)		129(286)	
Spinning	63	2815(635) **	3489(757) **	142(250)	**	165(344)	**
Weaving	82	3310(628) **	4132(746) **	131(183)	**	137(241)	**
Finishing	42	3053(760) *	3810(929) ^ns^	81(229)	*	88(309)	^ns^
Garment	52	2688(711) **	3350(858) *	121(162)	**	105(256)	*

^a^ Paired *t*-test for the before and after shift comparison; ^b^ Independent *t*-test for the cross-shift change (FEV1 and FVC) between textile and control, and between the respective departments and control; Significance level at *p*-value, ^ns^ = *p* ≥ 0.05; * *p* < 0.05; ** *p* < 0.001.

**Table 5 ijerph-17-02741-t005:** Multiple linear regression of cross-shift change FEV_1_ and FVC in relation to exposure status among textile workers and controls.

Variable	B	SE	β	*p*-Value	95% CI
**∆FEV1, R^2^adj = 0.105, *n* = 340**					
(Constant)	−0.143	0.058		0.014 *	−0.257–−0.029
Controls (0), Exposed (1)	0.090	0.024	0.206	<0.001 *	0.043–0.137
Day shift (0), Evening shift (1)	0.089	0.022	0.220	<0.001 *	0.046–0.132
Animals in the house (0/1)	−0.045	0.030	−0.080	0.132	−0.103–0.013
Use biomass fuel (0/1)	0.014	0.023	0.0334	0.530	−0.031–0.060
Female (0), Male (1)	−0.021	−0.021	−0.052	0.332	−0.063–0.021
Age, years	0.002	0.001	0.105	0.050	0.000–0.004
**∆FVC, R^2^adj = 0.107, n = 340**					
(Constant)	−0.119	0.076		0.009 *	−0.349–−0.050
Controls (0), Exposed (1)	0.097	0.031	0.169	0.002 *	0.035–0.159
Day shift (0), Evening shift (1)	0.124	0.029	0.234	<0.001 *	0.068–0.181
Animals in the house (0/1)	−0.083	0.039	−0.113	0.033 *	−0.159–−0.007
Use biomass fuel (0/1)	0.040	0.030	0.071	0.184	−0.019–0.099
Female (0), Male (1)	−0.051	0.028	−0.098	0.070	−0.107–0.004
Age, years	0.003	0.001	0.123	0.022 *	0.000–0.006

* Significant *p*-value; SE = Standard Error; R^2^ adj = Adjusted R square; B = Coefficient; CI = Confidence interval.

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
