# Peer review of "Reduced Cross-Shift Lung Function and Respiratory Symptoms among Integrated Textile Factory Workers in Ethiopia"

_ijerph, 2020, doi:10.3390/ijerph17082741_

Round 1

Reviewer 1 Report

The authors have conducted a cross-shift study on textile workers in Ethiopia. The issue is intersting and of relevance for this occupational group. The paper is well written and easy to read. The design is appropriate for hypothesis-building and accordingly the conclusions are cautious. The paper deserves publication in the Journal, although it needs some improvements previously.

  1. The format of the references list is not according the Journal requirements particularly journal referencing (which in addition ist not unified through the list), Reference 19 is lacking the journal.
  2. Figure 1 should be moved to the results sections.
  3. In line 186 the authors say “in the previous and part of this study” I think is better to say: “In a previous study (reference 28) we…” and indicate the timepoint of sampling (e.g in August 2017…). The results of sampling (lines 198-204) should be used for the discussion (see below) as explanatation for the differences between departements.
  4. Regarding the participant characteristics it would be informative to know the proportion of subjects with obstructive pattern in spirometry (FEV1/FVC<70%) among control and study group. The authors report that there are no reference values for Ethipian people, but wouldnt it be a possibility to use the GLI multi-ethnic reference values? This way they could report the proportion of subjects with values for FEV1 and FVC <LLN. At least, it should be discussed why they GLI reference values were not used.
  5. In the Methods section line 220 the authors report that in the logistic regression for respiratory symptoms they adjusted for biomass fuel use, animals living in the house, education and working hours per week: accordingly the report the same for table 2. For lines 265-273, Table 3 however, when comparing the different parts of the textile factory they adjusted for gender, education, job, work in other dusty workplaces, cook own food and animals 268 living in the house. I think the model used should be consistent through the whole analysis of complaints. All variables might be relevant: age, gender, animals at home, biomass fuel using, cooking own food (perhaps both biomass fuel and cooking correlate, then you can choose only one of them), work in previous dusty places, etc.
  6. Lines 265-273 pleas report 95%.CI for the OR reported here.
  7. Format of tables: please check that all notes are superscript both in the table as well as in the legend of the table.
  8. Discussion: the discussion needs to be much more elaborated:
    a. First of all, I would include here the data of dust sampling (now lines 198-204) when discussing the different levels of exposure in the different parts of the factory and the differences in spirometry accross departaments (323-325). More levele of detail i salso needed: did the ones with higher dust exposure had worse spirometries? Discuss why the levels of exposition differ that much.
    b. 325-329 here you need to provide hypothesis why do you think your results differ from other studies, (e.g. differences in levels of occupational hygiene, in study methods, etc.)
    c. An important part of the discussion drives on the issue of the use of personal sampling as a strength of the study, but sampling is not the focus of the paper.

Reviewer 2 Report

General

The author assessed respiratory symptoms and cross-shift changes in lung function of textile factory workers exposed to cotton dust compared with not exposed controls. The prevalence of chronic cough, chest tightness and breathlessness was higher in textile workers than in controls. Higher cross-shift reduction of FEV1 and FVC was detected textile workers. Substantial differences in the prevalence of symptoms and lung function changes were observed among the factory departments. The study appears carefully conducted and the text reads well. The conclusions are sufficiently prudent and the limitations of the study are underlined. The originality is limited since there are many investigations in the literature on respiratory symptoms and lung function in cotton dust workers with similar findings. However, there are some elements of novelty given by the assessment of airborne dust and endotoxin work exposure and by the evaluation of all the department of the factory. There are some issues to consider in the design of the study and in the interpretation of data.

Specific

It is well established that cross-shift changes in lung function are higher on the first day of working week (usually on Monday). It is not specified when spirometry was performed in cotton workers. The timing of lung function measurement is critical for the interpretation of the findings.
Apparently, symptoms and cross-shift lung function are not related each other. In addition, symptoms seem associated with personal dust levels, whereas cross-shift changes seem related mainly with endotoxin exposure. These probably represent the most original part of the paper. However, the relationship with work exposure was not properly addressed in the analysis neither adequately discussed. In addition, the reason why workers in garment department, i.e. those with the lowest dust and endotoxin work exposure, had a significant cross-shift reduction of FEV1 and FVC  should be explained.

Minor

Line 17: the meaning of ‘integrated’ is not clear in the abstract without reading the main manuscript.

Line 99: Probably, ‘lung function’ would be more appropriated than ‘reduction’.

Line 150: ‘1 cigarette a day for 1 year’ is a rather negligible smoking habit to consider an individual as a smoker. Was it intended 1 pack a day? Please, clarify.

Round 2

Reviewer 1 Report

I have reviewed the current revised manuscriot of "Reduced Cross-shift Lung Function and Respiratory Symptoms among Integrated Textile Factory Workers in Ethiopia". Although the authors have addressed most of my points, some of them still require more work. 1. Lines 173-174. The authors argue that GLI may not be approrpiate for lower socioecnomic backgrounds and refer to references 28-30. The authors should give some of reasons discussed /found in those references why is GLI innapropriate. 2. Lines 254-268 In this part, the model used seems to differ from the others. As far as reported here , age was not included. The authors use the acronym AOR, this one should be introduced in the methods part. 3. I had recommended to provide a more elaborated discussion. Besides moving the part on exposure to Endotoxin to discussion as recommended by me, the authors have added merely a single sentence to the discussion (öines 325-326). In my view, this is not enough. The authors should discuss more in depth this aspect: proved more inforamntion on the others studies and the differences. and draw explanations from it. Why do the authors mean that work shif and time of measurment can explain the differences? How do "work shift" and "time of measurement" differ between their study and the previous ones? Are these the only two points to discuss? 4. The reference list still not being uniform. Compare for example references 4 and 7 and 19. In Addition I am affraid, that the reference list still not following the journal requirements (for example the authors report the full Journal Name...)

Reviewer 2 Report

The authors adequately considered the remarks of the reviewer and modified the manuscript accordingly.

There are not further comments.

Author Response

Reviewer point: The authors adequately considered the remarks of the reviewer and modified the manuscript accordingly. There are not further comments.

Authors response: Thank you for the constructive comments and feedback you have given us.